# Doping in Sport—Attitudes of Physical Trainers Students Regarding the Use of Prohibited Substances Increasing Performance

**DOI:** 10.3390/ijerph20054574

**Published:** 2023-03-04

**Authors:** Magdalena Zmuda Palka, Monika Bigosińska, Matylda Siwek, Boryana Angelova-Igova, Dawid Konrad Mucha

**Affiliations:** 1Department of Humanities, Section of Pedagogy, Faculty of Physical Education and Sport, The University of the Physical Education in Krakow, 31-571 Krakow, Poland; 2Institute of Physical Culture, State University of Applied Sciences in Nowy Sącz, 33-300 Nowy Sącz, Poland; 3Department of Tourism and Regional Studies, Institute of Law, Economics and Administration, Pedagogical University of Krakow, 30-084 Krakow, Poland; 4National Sports Academy Vassil Levski, Philosophy and Sociology of Sport, 1700 Sophia, Bulgaria; 5Department of Medicine and Health Sciences, Andrzej Frycz Modrzewski Krakow University, 30-705 Kraków, Poland

**Keywords:** physical trainer, Poland, doping, performance-enhancing substances, recreational drug

## Abstract

Background: The popularity of using the advice of a personal trainer is increasing in Poland and currently most gyms offer the possibility of training under the supervision of a professional. Personal trainers present a multifaceted nature into physical activity and become their clients’ guides in achieving sporting goals. Physical trainers also work in sports clubs and supervise the training of people professionally involved in sport. Aim: Given the professional role that they play, this article aimed to analyze the knowledge and attitudes of personal trainers towards using prohibited measures to improve performance in sport, as well as counteraction measures. Methods: The study used a questionnaire created by the authors containing closed, semi-open, and open questions. Results: The results of the presented research indicate that most physical trainers and students educated in this field have a negative attitude towards the use of prohibited measures that increase performance but they noticed that doping was common in sport by 88.51% respondents. In the group of personal trainers, the majority (87.14%) admitted that good results in sport could be achieved without the use of doping. They stated that it was unfair (25%), contrary to the fair play principle—16%, while over 11% indicated this as cheating. Only 6% of people pointed out that it was legally prohibited and 3% that it was harmful. These results show that 10.13% of all respondents believe that the use of doping is a necessity to achieve very good results in sport. Conclusions: The availability of doping substances is statistically correlated with the question of persuading to use doping in both groups of trainers and students and some people justify the use of doping. The research proved that the personal trainers’ level of knowledge on doping is still insufficient.

## 1. Introduction

Contemporary sport in the 21st century is marked by the development of equipment and clothing with the constant improvement of products, the use of which can contribute to an achievement of the highest possible result. Other companies are introducing both dietary supplements and food products to strengthen the physical condition. However, it should be constantly taken into consideration that any improvements should be implemented due to ‘the spirit of sport’, which is defined as ‘the pursuit of human excellence through the dedicated perfection of each person’s natural talents’ [1]. All means of performance-enhancing substances (PES) that are placed on the World Anti-Doping Agency (WADA) Prohibited List are considered as doping. Due to the development of biotechnology and biomedicine, this list is constantly increasing as newer ways of substances emerge. Additionally, we deal with the so-called recreational drugs (social drugs), which are psychoactive drugs for recreational purposes (e.g., hashish, marihuana, heroin, ephedrine, or amphetamine) [2]. Other sports equipment and clothing are not discussed in this article.

The subject of doping has been repeatedly undertaken in scientific, medical, psychological, and social works of recent years. Previous research has analyzed various sports. Special attention has been paid to professional athletes and their opinion on doping substances. The research group most often consisted of competitors of various sports disciplines, such as cycling [3,4], judo [5], sailing [6], football [7,8], swimming [9], weightlifting [10], tennis [11], and Paralympians and athletes with disabilities [12]. What is more, some recent reports have dealt with a large research group that consisted of Polish athletes of 13 sports disciplines [13]. Research into the attitudes of coaches in various disciplines towards doping [14,15] shows that to the present, there have been no articles referring to personal trainers.

### 1.1. Hypothesis

Due to the lack of research related to the opinion of physical trainers on the use of doping, a decision was made to further explore this issue. The popularity of using personal trainers’ advice in Poland is increasing and most of the gyms offer the possibility of training under the supervision of a professional. The work of physical trainers is largely focused on the comprehensive care over a non-professional athlete. They introduce people who are not related to professional sport into the world of athletics. Personal trainers present a multifaceted nature into physical activity and become their clients’ guides in achieving sporting goals. Physical trainers also work in sports clubs and supervise the training of people professionally involved in sport. In our opinion, trainers who work both at gyms and sports clubs can be authorities in the selection of supportive measures or even doping. This raises questions—what attitudes do personal trainers have towards doping? Is the use of PES common in sport? Do they consider it to be cheating or do they consider it to be an individual matter for each person?

### 1.2. Purpose

Due to the professional role that they play, the aim of this article was the diagnosis of knowledge and attitudes of personal trainers towards using prohibited measures to improve performances in sport, as well as any counteraction. The results of this research could create a base for further studies related to the ways of transferring the knowledge of personal trainers about doping to their clients and their influence on them.

### 1.3. The Problem of Doping

According to the WADA, doping is contrary to the ‘sport spirit’, which is cultivated naturally, following the rules, without artificial improvements. Doping remains opposed to the ethical principles of sport and medicine [1]. Doping tests, for the first time, were organized by the International Olympic Committee (IOC) in the 1960s. A medical committee established by the IOC prepared rules for the detection of doping in sport. However, the commission’s work was questioned due to the poor detection of doping. With the development of biomedicine and biotechnology, new doping substances have appeared, and thus new tests for their detection [16]. Along with the changes taking place in the anti-doping sphere, the non-governmental organization World Anti-Doping was established in 1999. This independent and international organization has become an institution defending the principle of pure rivalry (play true). Furthermore, the WADA code, created in 2003, clearly defines the objectives of the World Anti-Doping Program [17,18].

For many athletes, doping is part of a difficult choice. On the one hand, it is forbidden in sport and generally considered immoral and harmful to health, while on the other hand, some athletes believe that success in sports appears or becomes possible only through the use of such substances [19]. In the era of the commercialization and medicalization of sport, results and new records are what counts. Sport aims to impress and arouse positive emotions. Telegenic sports are more profitable, and athletes representing popular sports disciplines earn more money. Contemporary sport is based on the media-sport-complex principle [20]. Therefore, athletes are required to constantly achieve high results, best records, and to always have a high level of athletic performance. The pressure of society, high expectations, and social pressures may lead to athletes succumbing to the ‘temptation to be on doping’. According to R. Merton’s theory of structural tensions, reaching for doping creates a discrepancy between values, such as prestige and popularity, which are easier to achieve after using prohibited substances [21,22]. It is also problematic that some competitors manage to effectively avoid control. In addition, information suggesting that it is not possible to win without the use of additional banned substances and that all athletes ‘take’ them, offers another motivation to the athlete to reach for doping.

Honesty, respect for rules, health, fun, joy, dedication, commitment, courage, community, and solidarity are among the values that are attributed to the ‘spirit of sport’. Therefore, supporting oneself with the use of prohibited substances is against the axiology of sport. The use of doping is connected with the non-compliance of the rules and principles, including the ideas of ‘fair play’ or ‘pure play’. The use of doping creates social inequalities among athletes. The assumptions of anti-doping activities are aimed at equalizing opportunities between sportspeople. Victory should not be decided by non-sport factors, but only by ‘natural’ skills shaped as a result of training, such as endurance, efficiency, and strength. Pharmacological or technological doping contributes to facilitating the crossing of the limits of human endurance [23], and the athlete’s body becomes less and less natural [24].

Attempts to test doping are difficult due to the athletes’ fear of revealing the truth. It is also problematic to estimate the violation of the anti-doping rules as the controls are performed at random. Similarly, it is hard to try to determine which sport is the most burdened with the risk of doping, as the number of tests in a particular sports discipline is different each year. The list of prohibited substances also changes year by year. Substances that were admitted last year may be completely banned in the next. As can be seen from the statistical analyses, there is an upward trend in the number of anti-doping tests carried out [25]. In Poland, in 2005, out of the 1460 anti-doping tests conducted, 23 infringements were detected, and in 2016, out of the 3282 analyses, doping was confirmed in 49 athletes [26].

## 2. Materials and Methods

### 2.1. Participants

The study comprised 154 students in the Master’s program of the University School of Physical Education in Krakow, Poland and was conducted in January 2019. At the time of the research, these people had five months remaining until completing their education, receiving a master’s degree in physical education with a specialization of a physical trainer. The group was divided into two subgroups—the first group (70 people) were people already working as a personal trainer. The second group (78 people) were non-working students, who were approaching completion of the Master’s program. The average age in the group of physical trainers was 24.6 ± 1.76, the average age of the students was 24.2 ± 1.53. The average age of both groups was 24.4. Regarding the respondents, they practiced a variety of sports. The most popular were football (30 people) and weight training (26 people). This was followed by jogging (11), swimming (10), downhill skiing (9), fitness (9), and volleyball (9). In addition, combat sports (6), tennis (6), individual calisthenics, handball, boxing, basketball, climbing, floorball, badminton, skating, sports gymnastics were also mentioned. There were 32 people who did not give an answer on this topic.

### 2.2. Research Design

The research was carried out using a study questionnaire created for the needs of this particular research, in which participants were asked to provide answers. In addition to the metrics, the survey included four closed questions, six semi-open questions, and one open question regarding the use of doping. The qualitative open-ended questionnaire was used to assess the experience and opinion of personal trainers. Out of all the survey questionnaires completed, six questionnaires were rejected with errors or large gaps, which resulted in the final result of 148 people. The following questions were asked: (1) Is doping common in sport, and is its use a necessity on the way to achieving sports success? (2) Is it possible to justify the use of doping in any way? (3) Is there easy access to prohibited stimulants and by whom is it distributed most often? (4) Have you ever been urged to take doping and in what situation? (5) Are there people who use doping among your friends? What prompts them to take prohibited stimulants? (6) What is doping used for? (7) Does reaching for doping have a positive or negative connotation, or maybe one should approach this phenomenon individually? (8) In which disciplines, in the opinion of the respondents, is the phenomenon of doping present most often, and in which one the least? (9) Are there organizations known to try to combat the use of doping in sport? The research project gained the approval of the Commission on Research Ethics at the Institute of Applied Psychology of the Jagiellonian University in Krakow (Opinion no. 54 20 September 2019).

### 2.3. Statistical Analysis

Based on the results, a statistical analysis was carried out using the Statistica 13 program. The data characteristics were presented as percentage values, means, and standard deviations. To evaluate the differences between the groups, the non-parametric Mann-Whitney U test was used. The relationship between the pairs of variables was calculated using the Spearman’s rank correlation coefficient. The level of *p* < 0.05 was accepted as the significance level. No statistically significant difference was found between the groups of students and trainers in the answers regarding the use of doping agents. The detailed structure of the answers is presented in Table 1.

## 3. Results

Almost all respondents from both groups noticed that doping was common in sport (88.51%). In the group of personal trainers, the majority (87.14%) admitted that good results in sport could be achieved without the use of doping. It should be noted that 12.86% of trainers answered this question negatively. This opinion may affect their professional work and contact with clients. These results show that 10.13% of all the respondents (including more than 5% among trainers) believe that the use of doping is a necessity to achieve very good results in sport.Most trainers (87.14%) are strongly opposed to doping. They stated that it was unfair (25%), contrary to the fair play principle—16%, while over 11% indicated this as cheating. A share of 8.43% of respondents admitted that doping could be justified in the case when an athlete was subjected to drug therapy, which was on the list of prohibited measures. Furthermore, a few people (3%) think that doping could be justified because everyone ‘takes it’. For 4% of respondents, doping could be justified if everyone could take it. When comparing both groups, it can be seen that opinions did not differ much between the two groups.In the research we were also interested in access to doping substances. As many as 40% of trainers were not able to answer this question. Of those, 35.71% answered that access was easy and 24.29% that it was difficult. According to the group of trainers, banned substances were most often distributed by traders for 65.71% of respondents and friends for 44.29% of respondents. It was less often, according to the respondents, that trainers at gyms were responsible for the distribution—24.29%; similarly, trainers at sports clubs—25.71% and sports activists—21.43%. According to 11.43% of respondents in both groups, doctors were indicated as responsible for the distribution of prohibited substances.Among the two research groups, the majority of respondents do not have friends who use doping (72.29%). For the trainers 65.71% and 78.21% of non-working people do not have friends who take doping or do not know about it. A share of 34.29% of trainers and 21.79% of students admitted that there was a person who used doping among their friends. In the group of trainers, not everyone answered this question. The main reasons for taking prohibited substances among the trainers’ friends included: an increase in muscle mass—17% and improvement of results—5%.A separate question was aimed at learning the motivation for reaching for doping, among all respondents. According to the majority (85.71%) of physical trainers, doping is used to speed up the results, overcome the natural barrier of human capabilities 66.75%, strive to be better 65.71%, and out of curiosity, which was indicated by 14.28% of respondents. Similar results were obtained in the second group, where 71.79%, for the acceleration of results, was the biggest motivation to reach for prohibited measures, the desire to be better than others was chosen by 55% of respondents, and overcoming human capabilities by 45% of respondents. On the other hand, 18.75% of students pointed to curiosity.In the next question, the opinion that doping is perceived negatively by physical trainers has once again been confirmed by 70% and as cheating by 78.57%. People who recognized that doping was an individual matter (7.14%) and that it was a necessity of modern times (4.28%) had fewer negative associations. It was observed that all those for whom the issue of doping was an individual matter and was a necessity of modern times knew a person who took doping. We assume that people who have not commented negatively on doping could have had contact with such substances in the past or do now.According to personal trainers, the sports disciplines in which doping plays the most important role include cycling—67%, weightlifting—61%, bodybuilding—50%, athletics—57%, and cross-country skiing—56%. According to the surveyed trainers, sport disciplines where doping is less important included chess—41%, table tennis—15%; ski jumping, football, and volleyball—13% each; as well as dancing—12%. The opinions of other respondents were similar.Among many anti-doping organizations listed in the questionnaire, the IOC—68.91% and WADA—68.91% are the most well-known. The WADA is slightly more known among trainers—78.57% then the IOC—75.71%. The Commission for Combating Doping in Sport is known by 31.42% of respondents, and UNESCO by 17.14%. Three people have not heard of any anti-doping organizations. When comparing both groups, it should be noted that the trainers knew many more organizations operating in the anti-doping field.

The availability of doping substances is statistically correlated with the question of persuading the use of doping in both groups of trainers and students. Similar dependencies were found in questions about the easy availability of these substances and persuading for the use of doping and the question of friends using doping. From this, it follows that these people had indirect contact, presumably even direct, with forbidden stimulants. In the group of trainers, there was a statistically significant positive correlation between the urge to use doping and the question of whether its use could be justified in any way. People who were offered to use doping sought a justification for taking it. Another dependence was also found between the question about the possibility of achieving good sports results without doping substances and the question about friends taking these substances and whether the examined person had been persuaded to use such support in their athletic performance.

## 4. Discussion

The results of the presented research indicate that most physical trainers and students educating in this field have a negative attitude towards using prohibited substances to increase performance. The majority perceives doping as inconsistent with the values of sport, they see it as cheating. Similar results were observed in the study by Morante-Sánchez and Zabala [14], which was conducted on technical staff members because both the research group from this study and ours consisted of non-athletes; this study seems important for us. Other studies also confirm that too many people doping in sport is a fraud [4]. Although the majority of respondents refer to doping in a negative manner, which distorts the sense of sport and gives unfair results, there were almost 13% of people among the physical trainers, for whom sports success could not be achieved without using prohibited substances. Nearly 13% of respondents stated that doping could be justified as it could be due to the use of drugs, or it was justified by the fact that all athletes use it. This, in turn, refers to the results of the Tangen and Breivik studies [27] in which they indicated that the athlete’s decision on doping depended on the competitors also taking prohibited support substances. If athletes thought that others reached for doping, they also used it. This creates a vicious circle, which was pointed to by Morante-Sánchez, Mateo-March, and Zabala [4].

For one-third of respondents, doping substances were easily available. It is comforting that nearly 28% of performance enhancers were hard to access. The rest of the people who did not know this subject indicated that they had never thought about using doping at home or among friends. According to our respondents, traders and acquaintances are responsible were responsible for their propagation. The results of our research do not confirm the results of other authors’ studies, which have shown that mainly managers [4,7,28] and trainers [29] were responsible for delivering doping. Forbidden doping substances were proposed to 11% of our respondents. Among the Spanish technical football team it was 5% and over 9% among trainers [14].

Among all respondents, as much as 27.7% of people knew someone who took or used to take prohibited supportive substances. However, none of our respondents admitted to having used doping. As reasons for reaching for doping, the athletes most often indicated the acceleration of results. Similar results are presented in the article by Striegel, Volkommer, and Dickuth [30]. Previous studies of other authors noted that obtaining measurable financial gains could be one of the goals of using prohibited substances [4,31], which has not been confirmed in our research. Among our respondents, some people stated that reaching for doping was an individual matter and that there was nothing wrong with it. Similar results appeared in the studies of Alaranta et al. [32] and Waddington et al. [8]. In our opinion, even if a few people view the matter of taking doping in positive terms or justify it in any way, it should raise doubts as to their further work with clients-sportspeople. Especially those who have recognized that doping was an individual matter or a necessity of modern times (14%), or who had friends who used forbidden substances to enhance performance. In our opinion, this small group of respondents trusts doping and probably knows how to use it; they may have used it before, still use it, or intend to use it.

The sports disciplines indicated most often in which doping is used include cycling, weightlifting, bodybuilding, athletics, and cross-country skiing. We notice here the dependence between the indications for specific disciplines and the information provided by the media. An example of long-term use of doping by one of the most titled cyclists, Lance Armstrong, has probably caused that cycling was indicated most often by the respondents. Similarly in the case of cross-country skiing, which enjoys great popularity in Poland thanks to Justyna Kowalczyk. The respondents were familiar with the notoriety of taking asthma medications by competitors of the aforementioned athlete. On the other hand, the disciplines that were least exposed to using doping by athletes included chess, table tennis, or ski jumping. Let us just add that the question was open, without suggestions of a particular sports discipline.

There is not enough knowledge about doping organizations among physical trainers. Similar gaps in knowledge about doping have been repeated by respondents described in other publications [14,33]. Morante-Sánchez and Zabala [14] clearly state that insufficient knowledge about doping exists among footballers and also people who are involved in the preparation of Spanish footballers. Following the postulates of the previous studies [3,4,7,13,32,34], we have a similar opinion that, in addition to carrying out controls, anti-doping programs for juvenile athletes should be implemented from an early age. Our postulates regarding doping are to increase control among athletes and also to carry out appropriate anti-doping programs among personal trainers who have a great impact on non-professional sportspeople. This could improve the quality of their work due to a bigger awareness and in regard to this, a greater care and restraint in using dietary supplements. It would also have an educational aspect by passing this knowledge onto athletes.

The limitations of the present study are the inclusion of a group of master’s students. It would be necessary to extend this study to personal trainers who have a lot of experience but did not necessarily graduate from a sport-related master’s study. Additionally, it would be possible to divide these trainers into those who work in sports clubs and gyms and those who are involved in different sports.

## 5. Conclusions

The research proved that doping is common in sport. More than half of those questioned have a negative attitude towards doping, including that it is unfair and cheating. They also admitted that good results in sport can be achieved without doping. The group of personal trainers had more often received offers to use prohibited substances. Some respondents considered the use of doping to be an individual matter or a necessity of modern times, and these were participants who knew a person who had used doping. The analysis of the results also demonstrates that there is not enough knowledge on doping and also organizations acting against the use of doping in sport. Some respondents were not even able to indicate a single institution of this kind.

## Figures and Tables

**Table 1 ijerph-20-04574-t001:** The use of doping agents in the studied groups.

	Trainers (n = 70)			Students (n = 78)			
	Yes (%)	No (%)	I Don’t Know (%)	Yes (%)	No (%)	I Don’t Know (%)	*p*
Is doping a phenomenon commonly found in sport?	88.57	11.43	0.00	88.46	8.97	1.28	0.866
Is it possible to achieve very good results in sport without using doping?	87.14	12.86	0.00	88.46	7.69	3.85	0.881
Can the use of doping agents be justified in any way?	11.43	77.14	11.43	14.10	74.36	7.69	0.404
Are doping measures readily available?	35.71	24.29	40.00	29.49	30.77	38.46	0.754
Have you ever been urged to use doping?	12.86	87.14	0.00	8.97	91.03	0.00	0.451
Has any of your friends taken any doping substances?	34.29	35.71	30.00	21.79	41.03	37.18	0.134

## Data Availability

All data generated or analyzed during the study are included in this article.

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
