# Peer review of "Doping in Sport—Attitudes of Physical Trainers Students Regarding the Use of Prohibited Substances Increasing Performance"

_ijerph, 2023, doi:10.3390/ijerph20054574_

Round 1

Reviewer 1 Report

Dear Authors and Editors,

Although the authors have made considerable efforts to develop this paper, however, I believe that there are many major issues in manuscript titled Doping in Sport – Attitudes of Physical Trainers Students Regarding the Use of Prohibited Substances Increasing Performance.

I think that the overall structure and writing of introduction part are not clear and well-aligned because it is not easy to catch what the research questions and strategies in this paper. Please clearly describe those things.

Although this paper dealt with interesting phenomena, this paper did not provide the part about the novelty of study So, it is very difficult for me to be sure that the research has an enough level of theoretical value and contribution.

Study size: Indicate if sample size was calculated. How?

Has the survey questionnaire been validated?

It is recommended to include a section indicating the design of the study as well as the code of ethics.

The results section is unclear.

References and citations do not match MDPI Reference List and Citations Style Guide

Author Response

Dear Reviewer and Editors,

Thank you for your review, we have addressed all the objections and tried to improve it to make it clearer and more readable. The research questions have appeared in the article so that it is easier to understand the purpose of the research at lines 82-84.

Regarding the novelty of the area, we have addressed this at lines 65-71. Our research indicates that there is a paucity of knowledge among personal trainers about doping. The specialization of the personal trainer in Poland is a very fashionable direction and many people are becoming personal trainers, working not only with athletes, but also with those people who want to take care of their health or who do sports only recreationally. This is why attitudes towards doping are so important, as personal trainers should categorically oppose the taking of doping by their charges. To date, no research has been carried out in this field.

The selection of the sample was purposive. It was important that those entering the personal trainer profession made their attitudes towards doping clear. According to the sampling calculator, we assumed that approximately 2,000 people would graduate with a rate of 56%, where the fraction size was 0.5 and the error max. 3%. The questionnaire was self-administered. It was previously verified with 30 students to ensure that it was clear and understandable, and then a few corrections were applied to the actual questionnaire. Obtaining bioethical approval was added to the article line 163-165.  The bibliography has been changed, according to the MDPI publisher.

Your sincerely

Reviewer 2 Report

Thank you for the opportunity to review the manuscript.  Overall it is well written.

It appears that the students were recruited from a University of Physical Education in Kraskow and were at near completion of the Masters program.  My concern is with possible perceived coercion given that at least one of the research faculty is from the university.  How was coercion minimized/eliminated?  What was the inclusion/exclusion criteria of the participants both coaches and students?  What were the demographics of the sample?  Were the participant students all in sport? and what was the breakdown numbers of the sport? 

The instrument administered was a developed questionnaire with quantitative and qualitative properties.  How was internal validity of the questionnaire achieved?  How were the qualitative data analyzed? 

The acronyms WADA and NGO appear in the manuscript early without the definition.  WADA's definition is presented later under line 86.  NGO is under line 94 without definition.  

Recommend to briefly described the limitations of the study.  

Author Response

Dear Reviewer and Editors,

Thank you for your honest review. We have corrected the errors and will try to answer the questions.

Well, the respondents are Master's students who completed the questionnaire in the presence of an outsider, without having any contact or classes with the authors of the article. There can be no compulsion to complete the questionnaire here. There were two criteria for inclusion a. there was a master's degree in a sport-related subject b. the respondents were of a similar age of approximately 24 years.

The article included information on which sports the respondents practised, as suggested by the Reviewer (line 151-155).

The questionnaire was previously verified on 30 students for clarity and understanding, and a few corrections were then applied to the actual questionnaire.

Qualitative surveys were categorized into the same responses and percentages were calculated.

Entire names and abbreviations were written correctly according to the order in lines 51-52 and 89-90.

The limitations were dissected at the end of the discussion in lines 360-364.

Thank you once again

Your sincerely

Reviewer 3 Report

Dear Authors,

thank you for the opportunity to review your manuscript entiled; Doping in Sport – Attitudes of Physical Trainers Students Regarding the Use of Prohibited Substances Increasing Performance.

You should start introducing references in the order of 1, not reference number 35 which appears first

It is necessary to reformulate the sentence when describing the results of other people's research. It is not necessary to list the research titles. Lines 62-68

The results should be briefly described with an emphasis on the most important while you can use part of the text in the discussion section

Author Response

Dear Reviewer and Editors,

Thank you for your positive review.

The references have been corrected according to the reviewer and journal guidelines.

In lines 58-68, sports disciplines and titles were deliberately presented to emphasise that our research is novel and there is little information on attitudes towards doping by coaches, particularly personal trainers.

These sentences have been deleted, in accordance with the reviewer's wishes:

As it results from the study, doping is not the worst evil for everyone, some people perceive prohibited substances enhancing performance as an indispensable element of modern times or consider it to be an individual matter of every athlete. Some people justify the use of doping.

The combination of knowledge and practice of various sports, universities, and the WADA organizations may in the future provide tangible benefits from this cooperation.

This fragment from the conclusions was moved to the discussion:

Our postulates regarding doping are to increase control among athletes, but also to carry out appropriate anti-doping programs among personal trainers who have a great impact on non-professional sportsmen. This could improve the quality of their work due to a bigger awareness and in regards to this, greater care and restraint in using dietary supplements. It would also have an educational aspect, bypassing this knowledge to athletes. 

In addition, conclusions were reduced to a minimum and the most interesting conclusions were listed.

Thank you once again

Your sincerely

Round 2

Reviewer 1 Report

Dear authors,
Thank you for your revised version.

Author Response

Dear Reviewer

Thank you for your review. We are grateful for your valuable comments.  

Best regards, 

Authors.

Reviewer 3 Report

Dear Authors, thank you for considering the suggestions to improve your manuscript.

I suggest you shorten the paragraph (lines 64-70)

„However, insufficiency in the number of publications regarding personal trainers’ attitudes to doping has been observed in studies related to the topic, which were conducted in Spain [14] and Poland [15]. A study presented in the article titled: Knowledge, attitudes, and beliefs of technical staff towards doping in Spanish football, conducted by Morante-Sánchez and Zabala [14], where the respondents were trainers, physical trainers, and the rest of the technical staff, is worth mentioning, as well as the Polish study titled: Knowledge and attitudes of sports coaches with regard to doping and counteracting doping in sport [15].“

Reading your results, they still need to be written more extensively, and I advise you to cut them down to only the significant ones, and in the discussion, you can comment and compare them with other research.

Author Response

Dear Reviewer

Thank you for your review. We are grateful for your valuable comments, we have incorporated the comments in our manuscript, especially in the results.

I suggest you shorten the paragraph (lines 64-70)

Research into the attitudes of coaches in various disciplines towards doping [14,15], shows that to the present, there have been no articles referring to personal trainers.

Reading your results, they still need to be written more extensively, and I advise you to cut them down to only the significant ones, and in the discussion, you can comment and compare them with other research.

Also, the results have been modified and sections removed. We focused on the most important
results.
1. Similar research results were obtained from studies of non-working students. According to the majority (88.46%), very good results in sport could be achieved without doping, 7.69% of respondents answered this question negatively, and 3.85% could not answer this question
2. Only 6% of people pointed out that it was legally prohibited and 3% that it was harmful. However, not all respondents gave an exhaustive answer to this question. The majority of non-working students in the share of 74.36% stated that the use of doping agents could not be justified because it caused the result to be unpredictable - 19%, it is cheating - 26% of people and only 12% of respondents answered that it was incompatible with the fair play principle.
However, not all respondents gave an exhaustive answer to this question. The majority of non-working students in the share of 74.36% stated that the use of doping agents could not be justified because it caused the result to be unpredictable - 19%, it is cheating - 26% of people and only 12% of respondents answered that it was incompatible with the fair play principle. Doping
should not be justified, as it is legally prohibited- 9%, for 4% doping was associated with a lack of ethics and was against the rules of the sport. Surprisingly, only 4% of respondents mentioned the harmfulness of doping. This question, both among trainers and students, indicates that the social values and the character of sport are more important for this group of respondents than
health aspects. According to 13% of non-working students, doping could be justified if it was related to the treatment of a disease or the use of appropriate drugs that were on the list of prohibited substances

3. Similar results were recorded among non-employed students.
Among the group of students, traffickers were also most often referred to as persons responsible for distributing prohibited substances - 56.41% and friends-43.59%. Trainers at gyms were mentioned by 23.08%, sports activists by 21.79%, and trainers at clubs by 15.38%, as well as doctors by 14.10%. A share of 12.82% chose the answer ’others‘ but they could not indicate who else could be responsible for distributing doping.
4. Respondents also mentioned that the reason for taking doping by their friends was faster regeneration after physical effort - 2%. In the group of students, friends take them because this is related to the increase of muscles - 10%, and for 5% the main factor of taking doping was the increase in performance.
5. Also, one respondent from the group of trainers stated that there was nothing wrong with doping. The results of the students’ and personal trainers’ opinions were once again similar. Doping is perceived by students as cheating-64.10% and has a negative impact - 51.28%. For 10.25% of students, doping was an individual matter. Furthermore, for 6.41%, doping was a necessity of the modern world. Again, it was noticed that people who replied that doping was an individual matter and a necessity knew people taking doping.
6. Studies of non-working students, where 57% of respondents listed cycling as a discipline in which doping played the most important role. Weightlifting - 48%, athletics - 47%, cross-country skiing - 35% and bodybuilding - 31% were also listed.
7. Chess was in the first place - 25% of answers, volleyball was indicated by 15% and gymnastics, ski jumping, table tennis were indicated by 13% of students.
8. Individual persons know the Anti-Doping Information Ambulatory (5.71%), International Sports Federations (7.14%), or the Association of Anti-Doping Organisations-7.14%. In the group of students, IOC is known by 61.53% and WADA by 58.97%. The Commission on Combating Doping in sports is known only by 17.5%, and UNESCO by merely 8.75% of respondents. Other organizations were marked by individuals.

Yours sincerely
